# Hamster Sperm Possess Functional Na^+^/Ca^2+^-Exchanger 1: Its Implication in Hyperactivation

**DOI:** 10.3390/ijms24108905

**Published:** 2023-05-17

**Authors:** Gen L. Takei, Yuhei Ogura, Yoshihiro Ujihara, Fubito Toyama, Keitaro Hayashi, Tomoe Fujita

**Affiliations:** 1Department of Pharmacology and Toxicology, Dokkyo Medical University, 880 Kitakobayashi, Mibu, Tochigi 321-0293, Japan; 2Department of Electrical and Mechanical Engineering, Graduate School of Engineering, Nagoya Institute of Technology, Nagoya 466-8555, Japan; 3School of Engineering, Utsunomiya University, Yoto 7-1-2, Utsunomiya 321-8585, Japan

**Keywords:** sperm capacitation, sperm hyperactivation, Na^+^/Ca^2+^ exchanger, NCX1

## Abstract

Previous studies demonstrated that hamster sperm hyperactivation is suppressed by extracellular Na^+^ by lowering intracellular Ca^2+^ levels, and Na^+^/Ca^2+^-exchanger (NCX) specific inhibitors canceled the suppressive effects of extracellular Na^+^. These results suggest the involvement of NCX in the regulation of hyperactivation. However, direct evidence of the presence and functionality of NCX in hamster spermatozoa is still lacking. This study aimed to reveal that NCX is present and is functional in hamster spermatozoa. First, NCX1 and NCX2 transcripts were detected via RNA-seq analyses of hamster testis mRNAs, but only the NCX1 protein was detected. Next, NCX activity was determined by measuring the Na^+^-dependent Ca^2+^ influx using the Ca^2+^ indicator Fura-2. The Na^+^-dependent Ca^2+^ influx was detected in hamster spermatozoa, notably in the tail region. The Na^+^-dependent Ca^2+^ influx was inhibited by the NCX inhibitor SEA0400 at NCX1-specific concentrations. NCX1 activity was reduced after 3 h of incubation in capacitating conditions. These results, together with authors’ previous study, showed that hamster spermatozoa possesses functional NCX1 and that its activity was downregulated upon capacitation to trigger hyperactivation. This is the first study to successfully reveal the presence of NCX1 and its physiological function as a hyperactivation brake.

## 1. Introduction

In mammalian reproduction, molecular components cause flagellar to “wave” in a side-to-side movement. This flagellar waving, or “beating”, drives spermatozoa through the female reproductive tract, which is vital for reproduction. The active beating of flagella is initiated upon ejaculation, referred to as motility “activation” [1]. Following activation, the sperm cells undergo critical biochemical and physiological changes, known as “capacitation”, which ultimately enables sperm to fertilize oocytes. Capacitated spermatozoa are enabled with specialized motility, allowing for increased bending referred to as “hyperactivation” [2]. This process is necessary for the spermatozoa to penetrate the viscoelastic oviductal mucus and cumulus matrix [3,4]. Studies using knockout models on defective hyperactivation demonstrated that these models fail to fertilize both in vivo and in vitro [5,6,7]. Further studies have shown that sperm treatments aimed at upregulating hyperactivation can improve in vitro fertilization rate. These studies include transient exposure to Ca^2+^ ionophore [8,9] or the transient starvation and replenishment of the energy substrate [10]. Based on the observations of these studies, it has been suggested that hyperactivation is not only physiologically important but that it is also an attractive target for improved assisted reproductive technologies (ART).

It is well established that an increase in intracellular Ca^2+^ levels [11], the cAMP/protein kinase A (PKA) pathway, and the increase in tyrosine phosphorylation downstream of PKA, are involved in the regulation of capacitation/hyperactivation [12,13,14,15]. However, the precise mechanisms describing how spermatozoa regulate hyperactivation remains uncertain.

In the previous study, we reported that Na^+^ homeostasis plays a crucial role in hamster sperm hyperactivation in a membrane potential-independent manner [16,17,18]. These previously obtained results suggested that the Na^+^/Ca^2+^ exchanger (NCX) functions as a “brake” in suppressing the expression of hyperactivated motility [16]. However, the presence and function of NCX in hamster sperm has not yet been reported. Therefore, detecting the NCX protein, determining the isoform of NCX present in hamster spermatozoa, and revealing whether NCX is functional in hamster spermatozoa is warranted.

The aim of the present study was to reveal that the NCX is indeed present and functional in hamster spermatozoa. The results obtained showed that hamster testes express mRNA comprising both NCX1 and NCX2, but that hamster spermatozoa express only the NCX1 protein but not NCX2. In addition, results highlighted that although NCX activity was detected in hamster spermatozoa, activity was hindered by the presence of the NCX inhibitor (SEA0400) at an NCX1-specific concentration. Furthermore, NCX activity declined upon capacitation, presumably via a downregulation of PIP_2_ level. This suggests that capacitation-induced suppression of the NCX1 activity is necessary to trigger hyperactivated motility.

## 2. Results

### 2.1. Detection of NCX1 in Hamster Spermatozoa

First, we confirmed that extracellular Na^+^ delays hyperactivation (Appendix A) and lowers intracellular Ca^2+^ level (Appendix A), as previously reported. These results suggest that NCX-like activity is involved in hamster sperm hyperactivation.

Next, contigs with high homology to NCX isoforms were searched for in the expression profile of hamster testis mRNAs obtained via RNA-seq analysis (Table 1). Based on hamster testes, contigs with high homology to NCX1 (isoform X8) and NCX2 but not NCX3 were found (e-value < 0.001; Table 1). The expression of these isoforms was confirmed through conventional RT-PCR (Figure 1A). To confirm the existence of NCX1 and NCX2 proteins in sperm cells, Western blotting was performed using specific antibodies (Figure 1B–D). Two monoclonal antibodies (D3F3H from Cell Signaling Technology, Inc. Beverly, MA, USA and R3F1 from Swant AG, Burgdorf, Switzerland) were used to detect NCX1, with both antibodies detecting three positively reacting bands (~110 kDa, ~70 kDa and ~55 kDa) from a hamster sperm microsomal protein sample, and the 110 kDa band corresponded with the expected molecular weight (Figure 1B, red arrowhead). To validate the specificity of the antibody used, we synthesized the peptide against R3F1 antibodies, and Western blotting was performed with the antibody preabsorbed with the peptide (Figure 1C). The result showed that the band of expected molecular size was diminished in the blot using absorbed antibodies (Figure 1C, red arrowhead), indicating the specificity of the R3F1 antibody. Further, the localization of NCX1 in spermatozoa was searched for through immunofluorescence experiments. However, this search failed because the antibodies did not function efficiently in terms of immunofluorescence.

To detect NCX2 in hamster sperm, a custom rabbit polyclonal antiserum was produced. A ~110 kDa protein was detected using this antiserum in hamster brain preparation, indicating that antiserum functioned correctly (indicated by the red arrow in Figure 1D). The lower two bands detected in brain preparation were also revealed in the preimmune serum, suggesting that these signals were not derived from NCX2. In testis preparation, ~50 kDa, ~37 kDa, and ~30 kDa bands were detected. However, these bands were also detected using the preimmune serum. In addition, ~100 kDa and ~30 kDa proteins were detected through NCX2 antiserum taken from sperm samples; these two bands were also detected using the preimmune serum. These results suggested that the signals detected in testes and sperm samples were not derived from NCX2 protein. Moreover, results indicated that hamster spermatozoa contain NCX1 but not NCX2 at the protein level.

### 2.2. NCX1 Is Functional in Hamster Spermatozoa

Following the detection of NCX1 in hamster spermatozoa, the next step was to determine whether the detected NCX1 was functional. Therefore, Na^+^-driven Ca^2+^ uptake (reverse mode of NCX) was measured using a Ca^2+^ fluorescent dye, Fura-2. After Na^+^ loading into spermatozoa occurred through monensin, Na^+^-dependent Ca^2+^ uptake by NCX was initiated by replacing the medium with Na^+^-free BSS to reverse the Na^+^ concentration gradient across plasma membrane. All measurements were carried out in the presence of ouabain to inhibit NKA so as to allow the Na^+^ gradient across the plasma membrane to reverse and thapsigargin and Mibefradil to inhibit SERCA and CatSper, respectively. All of these reagents were added in order to avoid Ca^2+^ influx/efflux through channels/transporters other than NCX. NCX activity was measured in the head, midpiece, and principal piece of the spermatozoa, with the region of interest (ROI) being set, as shown in Figure 2A. The representative changes of the intracellular Ca^2+^ level in these regions are shown in Figure 2B. Medium replacement was performed at 10 s (Figure 2B). Because there were significant noises in the fluorescence signal in response to the medium replacement (5–15 s in Figure 2B), NCX activity was analyzed five seconds after the medium replacement (15 s in Figure 2B). The changes of Ca^2+^ levels after the detected noises disappeared were extracted and enlarged in Figure 2C. Following noise disappearance, Fura2 fluorescence intensity ratios in all ROIs increased linearly with time for ~10 s, with the increase gradually slowing down thereafter (Figure 2B,C). NCX activity was defined as the slope of the curve during the 10 s, which increased linearly based on the assumption that the curve was piecewise-linear (Figure 2C).

Region-specific NCX activities are summarized in Figure 2D. NCX activity was measured in all three regions, being significantly higher in the midpiece and principal piece than in the head (Figure 2D).

To determine region-specific NCX activity in more detail, sperm cells were divided into nine regions, as shown in Appendix A, and region-specific NCX activities were measured. Although no significant differences were detected via the Tukey–Kramer test due to the number of groups, the results showed that the midpiece and the regions adjacent to the midpiece (3–7 in Appendix A) displayed higher NCX activity than the acrosomal region (1 and 2 in Appendix A) and tip of the principal piece (9 in Appendix A). These results suggest that NCX is present primarily in the tail (midpiece and principal piece) region, but there is very little—if any—present in the head region.

### 2.3. The Effect of NCX Inhibitors on NCX Activity

To confirm whether the measured Na^+^-dependent Ca^2+^ uptake was driven by NCX, the effect of a specific NCX inhibitor was tested on it (Figure 3). SEA0400 was added at 1 µmol/L and 50 µmol/L; this was done because 1 µmol/L SEA0400 is almost capable of completely inhibiting NCX1, while 50 µmol/L SEA0400 inhibits both NCX1 and NCX2 [19]. When hamster spermatozoa were treated with 1 µmol/L SEA0400, the Na^+^-driven Ca^2+^ uptake was significantly inhibited in the head, midpiece, and principal piece (Figure 3A–F). When the concentration of SEA0400 was raised to 50 µmol/L, the inhibition of the Na^+^-driven Ca^2+^ uptake was not significantly altered, except for that in the midpiece (Figure 3A–F). Although 50 µmol/L SEA0400 more significantly reduced NCX activity than the presence of 1 µmol/L SEA0400 in the midpiece (Figure 3E), the inhibition was not as significant as that observed between the control and 1 µmol/L SEA0400. These results suggested that the Na^+^-driven Ca^2+^ uptake measured in hamster spermatozoa was driven primarily by NCX1.

### 2.4. The NCX1 Activity Was Declined upon Capacitation

We previously suggested that the downregulation of NCX is necessary for hamster spermatozoa to trigger hyperactivation [16]. To investigate whether NCX1 activity changes upon capacitation, NCX1 activity was measured both before and after capacitation (Figure 4A). When spermatozoa were incubated in capacitating conditions for 3 h to achieve capacitation, NCX1 activity significantly declined when compared to that without incubation (0 h; Figure 4A). Since intracellular Ca^2+^ levels rise upon capacitation, and this may affect NCX activity measurements, we determined basal intracellular Ca^2+^ levels before NCX activity measurements using the Fura-2 ratio at 0 and 3 h to take basal Ca^2+^ levels into consideration (Figure 4B,C). Indeed, basal intracellular Ca^2+^ levels were increased in the midpiece and principal piece of spermatozoa incubated in mTALP for 3 h as opposed to those incubated for 0 h (*p* = 0.0007 and *p* = 0.0418, respectively (Figure 4B). Figure 4B was enlarged and overlaid with a dot plot, and these raw data are shown in Appendix A. However, these increases in intracellular Ca^2+^ levels via capacitation were diminished when observed in BSS medium after Na^+^-loading procedure (*p* = 0.346 in head, *p* = 0.138 in midpiece, and *p* = 0.138 in principal piece, Figure 4C). Thus, NCX1 activity significantly declined in the capacitated spermatozoa, even when NCX1 activity was calibrated through the basal Ca^2+^ level (Figure 4D). These results suggest the robust regulation of NCX1 activity during capacitation. NCX1 protein levels were unchanged throughout the capacitation period (Figure 4E,F). Any changes in the PKA- or protein kinase C (PKC)-mediated phosphorylation of NCX1 were tested for via immunoprecipitation using anti phospho-PKA or -PKC substrate antibodies. However, immunoprecipitating the NCX1 protein was not possible using these antibodies.

### 2.5. PIP_2_ Level Was Decreased upon Capacitation

To further investigate the regulatory mechanisms of the downregulation of NCX1 activity, the PIP_2_ levels of sperm plasma membrane were examined through flow cytometry analysis using anti-PIP_2_ antibodies. As shown in Figure 5A, a decrease in the population of PIP_2_-positive cells as well as an increase in the population of PIP_2_-negative cells upon capacitation was observed. Indeed, the percentage of PIP_2_-positive cells was slightly but significantly decreased upon capacitation (Figure 5B, *p* = 0.01). These results suggest that decrease in PIP_2_ levels upon capacitation was involved in NCX1 downregulation.

## 3. Discussion

Calcium ions play an essential role in sperm function, such as maturation, acrosome reaction, flagellar motility, and swimming behavior [1]. Voltage-gated Ca^2+^ channels and the sperm-specific Ca^2+^ channel (CatSper) are reported to be involved in sperm function [20]. The CatSper channel, in particular, forms a pivotal role in biological functioning, and has been extensively studied [21]. Meanwhile, the Ca^2+^ efflux/clearance system has not been as extensively studied. Among the few studies conducted, Wennemuth et al. (2003) were able to detect Ca^2+^ clearance activity, most likely via NCX, mitochondria uniporter, and Plasma-membrane Calcium ATPase (PMCA), in mouse sperm [22]. Among these, PMCA4 has been shown to be present and is required for hyperactivation and fertilization in mouse spermatozoa [23,24] NCX function in spermatozoa has also been reported; however, the existence of full-length NCX proteins has not yet been shown, leaving its physiological role uncertain [25,26,27].

In the previous study conducted by the authors, using hamster spermatozoa, it was revealed that: (i) an increase in extracellular Na^+^ concentration lowers intracellular Ca^2+^ level, leading to a delayed expression of hyperactivation, and (ii) the NCX inhibitors, SEA0400 and SN-6, increase the intracellular Ca^2+^ level, consequently extinguishing the hyperactivation delay through extracellular Na^+^ [16]. Although the existence of NCX proteins was not revealed in previous studies, results do strongly suggest that NCX is present in hamster spermatozoa, operating as a “brake” for hyperactivation. In addition, it was suggested that the downregulation of NCX is necessary to express hyperactivated motility. Indeed, results from the present study highlight that NCX1 is present in hamster spermatozoa, and that its activity is downregulated upon capacitation. These results, together with the authors’ previous study, indicate that NCX1 is present and function as a brake to suppress premature hyperactivation in hamster spermatozoa (Figure 6). To date, the present study is the first of its kind to successfully indicate that the NCX1 protein is present and functional in spermatozoa.

In the present study, we detected three positively reacting bands (110 kDa, 70 kDa, and 55 kDa) via Western blotting using an NCX1 antibody. Since NCX1 has 12 different alternatively spliced isoforms in a tissue-specific manner [28,29], these variations in the molecular weight of NCX1 may derive from such isoforms. Indeed, previous study detected 160, 120, and 70 kDa NCX1 proteins from cardiac sarcolemma, all of which displayed exchanging activity [30]. It is likely that at least the 110 kDa and 70 kDa bands of sperm preparation correspond with those functional NCX1 detected in the previous study.

Several studies have reported that there are “accelerators” of hyperactivation, such as Ca^2+^ influx and Calmodulin kinase II [31,32], Nitric oxide [33], H_2_O_2_ [34], and hormones [35]. By contrast, only a few studies have reported the suppressive factor of hyperactivation: one of these studies being the authors’ previous works [16,36]. For efficient in vivo fertilization, following ovulation occurring within the oviduct, spermatozoa need to trigger hyperactivation in a timely manner. To achieve this spatiotemporal regulation of hyperactivation, the balance between “brake” and “accelerator” is crucial. Therefore, this study should provide valuable information towards the elucidation of the spaciotemporal regulation of hyperactivation within the female reproductive tract.

In the present study, it is conclusively shown that NCX1, but not NCX2 or NCX3, is present and functioning in hamster spermatozoa. There exists a possibility that the K^+^-dependent Na^+^/Ca^2+^ exchanger (NCKX) is involved in Na^+^-dependent Ca^2+^ regulation and hyperactivation in hamster spermatozoa instead of NCX1. This is because NCKX has been shown to regulate the intracellular Ca^2+^ level and sperm motility in sea urchin spermatozoa [37] and is also expressed in mouse testes [38]. In our study, we detected 1 µM of SEA0400-robust Na^+^-dependent Ca^2+^ influx (Figure 3). Since 1 µmol/L SEA0400 has no effect on NCKXs [19], this SEA0400-robust Ca^2+^ may be caused by NCKXs. The finding that 50 µmol/L SEA0400 inhibited the Na^+^-dependent Ca^2+^ uptake activity more intensely than that of 1 µmol/L in the midpiece suggests that SEA0400 affected mitochondrial transporters secondarily, because mitochondria are involved in the regulation of Ca^2+^ homeostasis and are restricted in the midpiece in spermatozoa.

It is not fully elucidated how spermatozoa downregulate NCX1 activity upon capacitation to release the hyperactivation “brake”. Since NCX1 activity is regulated by the PIP_2_ level on the plasma membrane [39], a decrease in PIP_2_ levels of sperm plasma membrane upon capacitation should account for the downregulated NCX1 activity, at least in part, although the extent of decrease was rather small. Other explanatory possibilities, such as the proteolysis or degradation of NCX1 or the phosphorylation of NCX1 by PKA or PKC [40,41], are not likely because of the following reasons. (1) The amount of NCX1 was not changed throughout the capacitation period. (2) Immunoprecipitations of NCX1 through anti-phospho-PKA or -PKC substrate antibodies were not possible. (3) The effect of phosphorylation by PKA or PKC on NCX1 is controversial and is still under debate [42]. There is a possibility that NCX1 is translocated from plasma membrane to intracellular vesicles during capacitation, but it was impossible to observe any change in NCX1 localization because the antibodies used in this study did not function effectively in immunofluorescence experiments. Another possibility is that NCX1 activity is regulated through the palmitoylation of NCX1 [43]. Further studies are needed to fully elucidate the regulatory mechanisms of how hamster spermatozoa downregulate PIP_2_ levels and NCX1 activity upon capacitation.

In conclusion, present findings show that NCX1 is present in hamster spermatozoa; furthermore, it functions as a hyperactivation “brake”. These findings can be used as the basis for further studies in order to effectively determine and understand how hamster spermatozoa downregulate NCX1 activity. This will assist in the better understanding of the regulatory mechanism and how mammalian spermatozoa develop hyperactivated motility over time. Furthermore, clarification on the regulatory mechanisms of hyperactivation can assist researchers in improving ART.

## 4. Materials and Methods

### 4.1. Reagents and Solutions

The acid buffering agent, N-2-Hydroxyethylpiperazine-N’-2-Ethanesulfonic Acid (HEPES), was purchased from Dojindo (Kumamoto, Japan). Bovine serum albumin (BSA) Fraction V was purchased from Merk KGaA (Darmstadt, Germany). The antibodies used against NCX1 were purchased from Cell Signaling Technology (D3F3H, Beverley, MA, USA) and Swant (R3F1, Bellinzona, Switzerland). The custom rabbit anti-NCX2 antiserum was obtained by immunizing rabbits with the peptide (CIGAEGDPPKSIELD) conjugated with Keyhole Limpet Hemocyanin (Merck KgaA, Darmstadt, Germany). Ouabain was purchased from Sigma Aldrich (St. Louis, MO, USA). The peptide sequence (DKQPLTSKEEEERRIAEMGRPILGEHTKLEVIIEESYEFKSTVDKLIKKTNLAL) against the R3F1 antibody for the blocking experiment was obtained from a previous study [44], and the peptide was synthesized by Merck KGaA (Darmstadt, Germany). Unless otherwise indicated, all other chemicals were purchased from Wako Pure Chemical (Osaka, Japan).

In the present study, a modified Tyrode’s albumin lactate pyruvate medium (mTALP) was used to induce hyperactivation in hamsters. This was carried out in accordance with a previously conducted study [16]. The composition of mTALP was conducted as follows: 101 mmol/L NaCl, 2.68 mmol/L KCl, 0.36 mmol/L NaH_2_PO_4_, 1.8 mmol/L CaCl_2_, 0.49 mmol/L MgCl_2_, 35.7 mmol/L NaHCO_3_, 4.5 mmol/L glucose, 1 mmol/L sodium pyruvate, 9.0 mmol/L lactic acid, 0.5 mmol/L hypotaurine, 0.05 mmol/L (-)epinephrine, 0.2 mmol/L sodium taurocholate, 5.26 μmol/L sodium metabisulfite, 0.1 mmol/L EDTA, 0.05% (*w*/*v*) penicillin G, 0.05% (*w*/*v*) streptomycin sulfate, and 15 mg/mL bovine serum albumin (BSA).

Standard Tyrode’s solution was used to load Fura-2, consisting of 140 mmol/L NaCl, 5.4 mmol/L KCl, 1.8 mmol/L CaCl_2_, 0.5 mmol/L MgCl_2_, 0.33 mmol/L NaH_2_PO_4_, 11 mmol/L glucose, and 5 mmol/L HEPES-NaOH (pH 7.4). A normal balanced salt solution (BSS) was used for NCX activity measurements, comprising 10 mmol/L Hepes/Tris (pH 7.4), 146 mmol/L NaCl, 4 mmol/L KCl, 2 mmol/L MgCl_2_, 0.1 mmol/L CaCl_2_, 10 mmol/L glucose, and 0.1% BSA.

### 4.2. Experimental Animals and Spermatozoa Collection

Spermatozoa were obtained from the cauda epididymis of sexually mature golden hamsters (*Mesocricetus auratus*). Study hamsters were euthanized using an overdose of isoflurane; the cauda epididymis was then isolated through dissection. The cauda epididymis was pierced with a 23-gauge needle, and the spermatozoa were then gently squeezed out from the epididymis. Collected spermatozoa were utilized as described in below sub-sections (Section 4.5, Section 4.6, Section 4.7 and Section 4.8). All experimental animals were kept and used in accordance with the guidelines of the Dokkyo Medical University and Nagoya Institute of Technology. All experiments conducted on animals were approved by the Animal Care and Use Committee of Dokkyo Medical University (Experimental Permission Number: 0996) and the Institutional Review Board of Animal Care at Nagoya Institute of Technology (Experimental Permission Number: 2021002).

### 4.3. RNA Seq Analysis

The total required hamster testis mRNA for the study was extracted using the RNeasy plus Universal Mini kit (Qiagen, Hilden, Germany). The quality of extracted RNA was evaluated using an Agilent 2100 Bioanalyzer (Agilent Technology, Santa Clara, CA, USA). Subsequently, paired end sequencing (101 bp read × 2) was performed by Illumina HiSeq 2500, as previously reported [45]. After base calling and chastity filtering, raw read data (41,976 Mbases; 415,600,252 reads; % of high-quality base (≥Q30): 92.51; Mean quality score: 36.06) were produced. Sequencing was performed by Hokkaido System Science Co., Ltd. (Sapporo, Japan). The raw reads were filtered to remove reads with adaptors and low-quality sequences (reads with unknown sequences ‘N’). Finally, 405,412,592 clean reads were yielded for the de novo assembly. These clean reads were assembled de novo using Trinity software (version 2.3.1) and annotated using NCBI non-redundant (nr) protein database.

### 4.4. PCR

The extraction of the RNA from hamster testes was conducted as described above (Section 4.3). The cDNA synthesis was performed using the Reva Tra Ace kit (Toyobo, Osaka, Japan). The sequences of hamster NCX1 and NCX2 were obtained from NCBI golden hamster RefSeq database. Specific primers were designed by primer3 software (http://bioinfo.ut.ee/primer3-0.4.0/ accessed on 6 November 2017). The sequences of forward and reverse primers were 5′-GTGGCTGAAAATGACCCAGT-3′ and 5′-CCGGGTTTGAAGATCACAGT-3′ for NCX1 and 5′-TCTGGAAGCCTAACCCAATG-3′ and 5′-ACCTTCCAGAACACCGTCAG-3′ for NCX2, respectively. PCR was performed using a Quick Taq HS DyeMix (TOYOBO, Osaka, Japan) for 35 cycles. The PCR products were separated by 1.5% (*w*/*v*) agarose gel, visualized by ethidium bromide, and captured by a FAS-IV ultraviolet transilluminator (Nippon Genetics Co., Ltd., Tokyo, Japan).

### 4.5. Microsome Preparation

Microsomal preparations were used for Western blotting to concentrate the membrane fraction of spermatozoa to detect NCX [17]. The spermatozoa, as collected above (Section 4.2), were washed with ice-cold PBS consisting of 137 mmol/L NaCl, 2.7 mmol/L KCl, 81 mmol/L Na_2_HPO_4_, and 14.7 mmol/L KH_2_PO_4_, at pH 7.4, and were resuspended in homogenization solution consisting of 250 mmol/L sucrose, 1 mmol/L EDTA, and 30 mmol/L histidine (pH 7.2 adjusted by adding Tris). Then, spermatozoa were homogenized by Polytron PT1300D (Kinematica, Lucerne, Switzerland) at 10,000 rpm on ice. Homogenates were centrifuged at 6800× *g* at 4 °C for 10 min. The supernatant was saved, and the resultant pellet was resuspended in the homogenization solution and centrifuged again as above. The supernatants from 2 centrifugations were combined and centrifuged at 45,000× *g* at 4 °C for 45 min. The pellet was resuspended in imidazole solution consisting of 1 mmol/L imidazole and 1 mmol/L Na-free EDTA (pH 7.4 adjusted by Tris), homogenized and centrifuged again at 45,000× *g* at 4 °C for 45 min. The pellet was resuspended in imidazole solution and stored at −80 °C until use. The protein concentration was determined using the Pierce BCA Protein Assay kit (Thermo Fisher Scientific, Waltham, MA, USA).

### 4.6. SDS-PAGE and Western Blotting

SDS-PAGE and Western blotting were performed as previously reported [17], with some modifications. Microsomal proteins, as prepared above (Section 4.5), were mixed with a sample buffer consisting of (final) 2% (*w*/*v*) sodium dodecyl sulfate, 25% (*v*/*v*) glycerol, 1% (*v*/*v*) β-mercaptoethanol, 0.01% (*w*/*v*) Bromophenol blue, and 62.5 mmol/L Tris-HCl pH 6.8 and boiled for 10 min at 100 °C. The boiled sample was electrophoresed in 8% (*w*/*v*) polyacrylamide separating gels and transferred to a polyvinylidene fluoride membrane (Immobilon-P, Millipore, Bedford, MA, USA). The blotted membrane was blocked with 4% BSA in TBS, containing 0.1% Tween 20 (TBST) for 1 h at RT. After blocking, the membrane was washed 3 times with TBST and incubated with an anti-NCX1 mouse monoclonal antibody (1:1000 dilution for both antibody) or an anti-NCX2 antiserum (1:5000 dilution) overnight at 4 °C. The membrane was then washed 3 times with TBST and incubated with a secondary antibody (anti-mouse IgG, HRP-linked antibody, Cell Signaling Technology cat# 7076 or anti-rabbit IgG, HRP-linked antibody, Cell Signaling Technology, cat # 7074, 1:1000 dilution for both) for 1 h at RT. After washing 3 times with TBST, the immunoreactive proteins were detected using SuperSignalTM West Pico Plus chemiluminescent substrate (Thermo Fisher Scientific). Western blots of three independent experiments were evaluated by densitometry using Image J software. The amount of total protein in each lane was set to 4 µg. Normalization using loading control proteins was not performed because the appropriate loading control protein of microsomal preparation was not known.

### 4.7. Measurement of NCX Activity

The NCX activity of sperm cells was examined using the fluorescent dye Fura-2, as reported previously [46], with some modifications. Cauda epididymal spermatozoa were suspended in an mTALP medium and incubated in a CO_2_ incubator (37 °C, 5% CO_2_, and 95% air) for 10 min (uncapacitated sperm) or 3 h (capacitated sperm). Aliquots of sperm suspension were then placed onto a concanavalin A-coated glass-bottomed dish (Matsunami Glass, Japan) and incubated at RT for 30 min to let the spermatozoa attach to the glass surface. The glass-attached spermatozoa were loaded with 10 µmol/L Fura-2 acetoxymethyl ester (Fura-2 AM) for 30 min at 37 °C in standard Tyrode’s solution. Fura-2-loaded cells were alternately excited at 340 nm and 380 nm using an LED illuminator for Fura-2 (pE-340fura, CoolLED, Andover, UK), while coupled to an inverted microscope IX73 (Olympus, Tokyo, Japan) with a water-immersed 40× objective lens (UApo N 340 40×/1.15NA W, Olympus) [47]. The Fura-2 fluorescent signal was recorded via an ORCA-Flash 4.0 (Hamamatsu Photonics, Hamamatsu, Japan) and analyzed using a ratiometric fluorescence method with MetaMorph software (Version 7.8.0.0; Molecular Devices, Sunnyvale, CA, USA). Na^+^-dependent Ca^2+^ uptake into cells was measured using a modification of the previously described method [46]. Fura-2-loaded spermatozoa were loaded with Na^+^ by incubation at 37 °C for 15 min in BSS containing 1 mmol/L ouabain, 1 mmol/L thapsigargin 1 mmol/L monensin, and 40 µmol/L Mibefradil. Ca^2+^ uptake was then initiated by switching the BSS medium to Na^+^-free BSS containing choline chloride, which was supplemented with 1 mmol/L ouabain, 1 mmol/L thapsigargin, and 40 µmol/L Mibefradil. We defined the Na^+^-dependent Ca^2+^ uptake, which is robust to these inhibitors but is sensitive to SEA0400 as NCX activity.

### 4.8. Analysis of PIP_2_ Levels by Flow Cytometry

Flow cytometry analysis was performed as previously described, with modifications [17]. Spermatozoa, as collected above (Section 4.2), were incubated in mTALP medium for 10 min or 3 h and were fixed by 4% paraformaldehyde in PBS for 10 min at RT. Fixed spermatozoa were centrifuged (370× *g* at 4 °C for 10 min) and washed with 0.2% BSA in PBS 3 times and were incubated with anti-PIP_2_ monoclonal antibody (2C11, Abcam, cat# ab11039, 1:250 dilution in 0.2% BSA in PBS) for 1 h at 4°C with continuous rotation. Then, spermatozoa were washed twice with 0.2% BSA in PBS and incubated with Alexa Fluor 488-conjugated goat anti-mouse IgM mu chain antibody (Abcam, cat# ab150121, 1:2000 dilution) for 1 h at 4 °C with continuous rotation. Then, spermatozoa were washed once, and the fluorescence was measured using a FACS Calibur flow cytometer (BD Biosciences, Franklin Lakes, USA) and analyzed using CellQuest Pro software (BD Biosciences). The Alexa Fluor 488-positive cells were considered PIP_2_-positive, and 15,000 cells were analyzed in each condition.

### 4.9. Statistical Analysis

Statistical analysis was performed via a one-way ANOVA and the Tukey–Kramer post hoc test, as shown in Figure 2D and Figure 3D–F. Results in Figure 4A–D,F were statistically analyzed through an (unpaired) Student’s *t*-test. Results in Figure 5B were analyzed by paired *t*-test. All the statistical analyses were performed using R-3.5.3 software. A *p*-value of <0.05 was considered statistically significant.

## Figures and Tables

**Figure 1 ijms-24-08905-f001:**
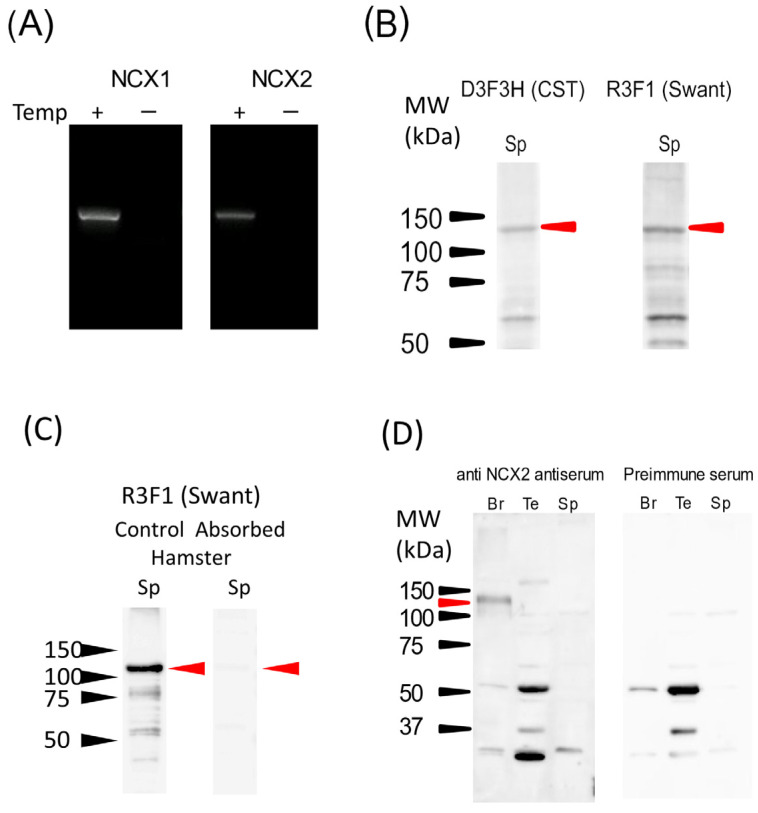
The expression of NCXs in hamster spermatozoa. The expression of NCX1 and NCX2 was examined via RT–PCR using hamster testis total RNA (**A**), and Western blotting using microsomal protein preparations from hamster spermatozoa (**B**–**D**). (**A**) The expression of NCX1 and NCX2 mRNA in hamster testes was investigated using conventional RT–PCR. PCR products without templates were used as the negative control. (**B**) The expression of NCX1 protein in hamster spermatozoa was investigated via Western blotting using two different antibodies (D3F3H from CST and R3F1 from Swant) against NCX1. (**C**) The R3F1 antibody was preabsorbed by blocking peptide, and Western blotting was performed to validate the specificity of the antibody. (**D**) The expression of NCX2 protein in hamster spermatozoa and testes were investigated using custom rabbit anti–NCX2 antiserum. Hamster brain samples were used as a positive control. The results of antiserum (left) and preimmune serum (right) are shown side by side. The amount of protein loaded was 10 µg/well. Red arrowheads indicate a positive band. The molecular weights in kDa are shown on the left side.

**Figure 2 ijms-24-08905-f002:**
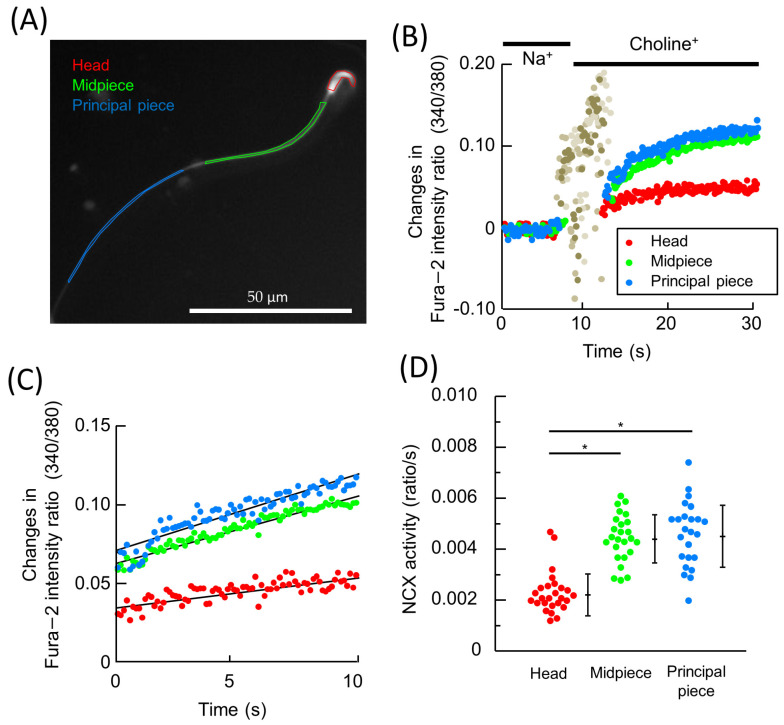
The region-specific measurement of NCX activity in hamster. The schematics of how NCX activity measurement was carried out (**A**–**C**) are shown, as well as the results of the measurements (**D**). The region of interest (ROI) was set as (**A**). (**B**): Na^+^-dependent Ca^2+^ influx was initiated via medium replacement (10–15 s), with the change in intracellular Ca^2+^ levels monitored by the ratio of Fura2. (**C**) The ratios of Fura2 in (**B**) 15 s after medium replacement were extracted and enlarged. The slope of each signal was defined as NCX activity. (**D**) The measured NCX activity (ratio/s) in head, midpiece, and principal piece are represented by a dot plot. Bars on the right represent mean ± S.D. A total of twenty-nine sperm cells from five individuals were assessed. Asterisks indicate significant differences among them (*p* < 0.05).

**Figure 3 ijms-24-08905-f003:**
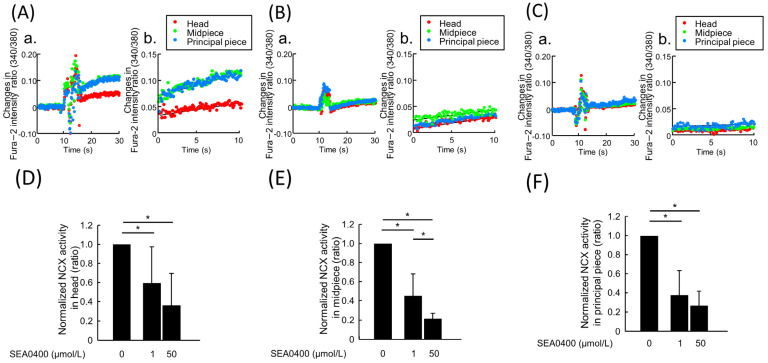
The effect of NCX inhibitor SEA0400 on Na^+^-dependent Ca^2+^ uptake by hamster spermatozoa. The effects of NCX inhibitor, SEA0400, on the Na^+^-dependent Ca^2+^ uptake were investigated. (**A**–**C**) Both overall time-course changes (**a**) and 15–25 s after the medium replacement (**b**) of Fura-2 ratio in the presence of 0 (**A**), 1 (**B**), and 50 µM (**C**) of SEA0400 were shown. (**D**–**F**) The measured NCX activity in head (**D**), midpiece (**E**), and principal piece (**F**) were normalized by those with an absence of SEA0400 and is represented by bar chart. Data are represented as mean ± S. D. Data were analyzed via a one-way ANOVA and the Tukey–Kramer post hoc test. Asterisks indicate significant differences among results (*p* < 0.05). A total of 26 cells was used to serve as the control, 30 cells for 1 µmol/L SEA0400, and 29 cells for 50 µmol/L SEA0400; all samples analyzed were taken from three individuals.

**Figure 4 ijms-24-08905-f004:**
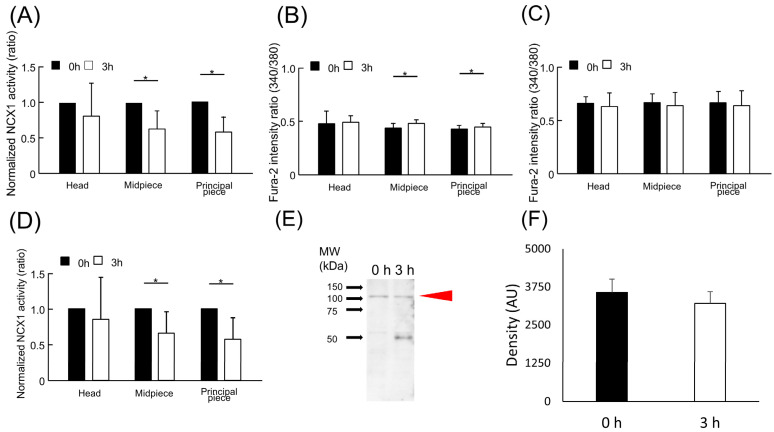
Capacitation-dependent change of NCX1 activity. (**A**) The change of NCX1 activity upon capacitation was investigated by measuring the NCX1 activity before (0 h) and 3 h after incubation (3 h) in capacitating conditions. (**B**,**C**) Basal intracellular Ca^2+^ levels of sperm before medium replacement were evaluated by Fura-2 ratio in the mTALP medium (**B**) or in the BSS medium (**C**) to confirm whether capacitation changed basal intracellular Ca^2+^ level during NCX activity measurement. (**D**) The NCX1 activity values (ratio/s) were calibrated by dividing them via the basal Fura-2 ratio in BSS medium (see (**C**)) and were normalized via the respective NCX1 activity values at 0 h in each region. Data are expressed as mean ± S.D. Data were analyzed using an (unpaired) Student’s *t*-test. Asterisks indicate significant differences among results (*p* < 0.05). A total of 29 cells for 0 h and 25 cells for 3 h from 5 individuals were analyzed in (**A**,**C**,**D**), and a total of 28 cells for 0 h and 22 cells for 3 h is analyzed in (**B**). (**E**) Protein levels of NCX1 before (0 h) and after (3 h) capacitation were investigated by Western blotting using microsomal preparations as a sample. The amount of protein loaded was 4 µg/well. Red arrowheads indicate a positive band. The molecular weights are shown on the left side. (**F**) The densitometry analysis of NCX1 bands was performed using Image J software for the comparison of NCX1 levels between 0 and 3 h. There was no significant difference between the results (*p* = 0.345). Data are expressed as mean ± S.D. n = 3.

**Figure 5 ijms-24-08905-f005:**
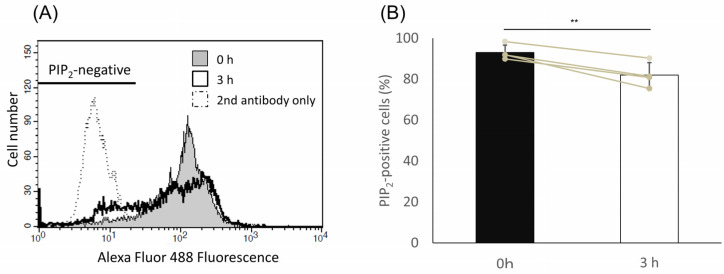
PIP_2_ level was decreased during capacitation. Hamster spermatozoa were incubated in capacitating condition for 0 h or 3 h, then the cells were fixed and labeled with anti-PIP_2_ antibodies and Alexa Fluor 488-conjugated secondary antibodies. The labeled cells were analyzed through flow cytometry to determine sperm PIP_2_ levels. (**A**) Typical histograms were represented. The histogram with dotted line represents the negative control (secondary antibody only). A bar on the top left represents the population of PIP_2_-negative cells. (**B**) Percentages of PIP_2_-positive cells of four individuals, as analyzed in (**A**), and are represented by bar charts and dot plots. Data are expressed as mean ± S.D. Asterisks indicate the significant difference between results (*p* = 0.01).

**Figure 6 ijms-24-08905-f006:**
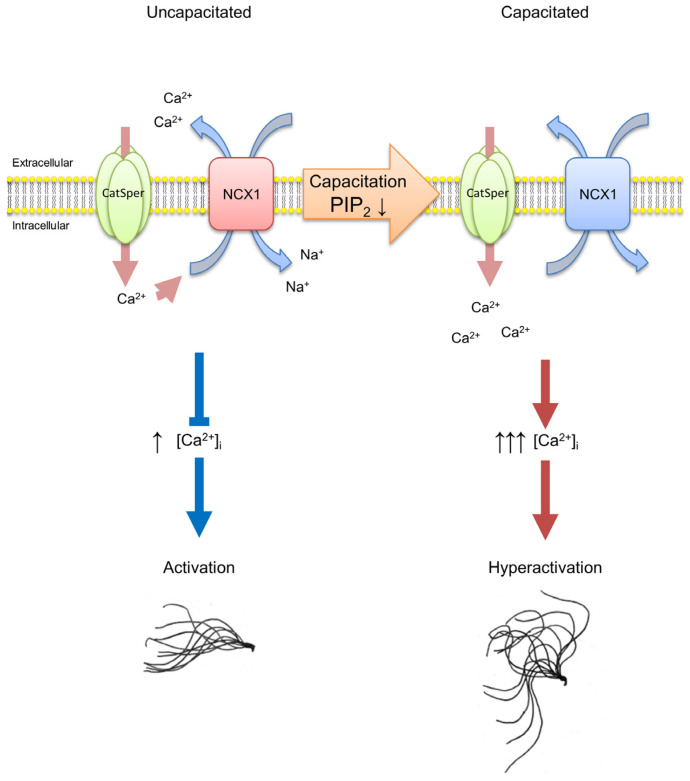
Schematics of hamster sperm hyperactivation regulation by NCX1. In an uncapacitated spermatozoa, NCX1 is active and extrudes intracellular Ca^2+^, which it presumably entered through the CatSper channel, thereby suppressing hyperactivation and remaining in an activation state. In capacitated spermatozoa, NCX1 activity is downregulated by a decrease in PIP_2_ levels and inactive. Thus, Ca^2+^ ions entered through the CatSper channel are not extruded. This causes an increase in intracellular Ca^2+^ levels ([Ca^2+^]_i_), leading to an expression of hyperactivated motility.

**Table 1 ijms-24-08905-t001:** NCX isoforms detected by RNA-seq analyses from hamster testes. NCX genes detected by RNA-seq analysis and bit score, e-value, % of identity, TPM (transcripts per kilobase million) and FPKM (fragments per kilobase of exon per million mapped reads) values were summarized.

Genes	Sequence ID	Species	Bit Score	e-Value	Identical (%)	TPM	FPKM
*NCX1*	XP_005077303.1	Mesocricetus auratus	1497.6	0	100	0.27	0.25
*NCX1*	XP_014420366.1	Camelus ferus	193.4	5.07 × 10^−62^	98.97	0.52	0.47
*NCX2*	XP_013853997.1	Sus scrofa	386.3	7.85 × 10^−131^	100	1.82	1.65

## Data Availability

The datasets generated during and/or analyzed during the current study are not publicly available but are available from the corresponding author upon reasonable request.

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
