# Peer review of "Hamster Sperm Possess Functional Na+/Ca2+-Exchanger 1: Its Implication in Hyperactivation"

_ijms, 2023, doi:10.3390/ijms24108905_

Round 1

Reviewer 1 Report

Some minor issues should be addressed before this work qualifies for publication.

1.      Figures 3D-E: What do the bars represent?

2.      Figure 4B: I doubt whether there are significant differences between 0h and 3h in the midpiece and principal piece. Could the authors provide the raw data? Also, since Ca2+ levels change rapidly, perhaps the authors could measure the Ca2+ level at another time point to observe a more robust increase.

3.      Figure 4E: a control protein should also be measured, and a quantification result is necessary since it is difficult to determine whether the band intensities have changed with the naked eye.

4.      Line 304-308: If mitochondrial Ca2+ transportation is also involved in this process, should the authors also block mitochondrial Ca2+ transporters when measuring NCX1 activity in Figures 2 and 3??

Reviewer 2 Report

The authors provided an exhaustive and well-designed study regarding the presence and physiological function of NCX1 as a brake in hyperactivation motility. 

1. The authors addressed the research question regarding functional aspects of sperm motility and in particular the mechanism controlling the hyperactivation.
2. In my opinion the topic is very interesting because we do not have completely understood alls per physiology and could represent a base for further clinical research also in other species as also human being.
3. The current manuscript add a new insight on presence and function of NCX1 in hamster’s sperm.
4. The authors conduced a well-designed study, which could be published as it is.
5. Conclusions are supported by evidences and arguments presented.
6. References are appropriate.
7. Figures and tables are explicative enough.

Author Response

We would like to express our thankfulness for sparing your time to review our manuscript. We appreciate your favorable comments on our study. We do not provide a point-by-point response because all the comments were favorable and no improvement was requested.

Round 2

Reviewer 1 Report

The authors have addressed all my concerns. I believe it is qualified for publication.

Author Response

We thank reviewer 1's for sparing his/her time to review our manuscript. We are happy to hear that we could addressed all his/her concerns.